

# The detection of trans gene fragments of hEPO in gene doping model mice by Taqman qPCR assay

Kai Aoki[1,*], Takehito Sugasawa[2,*], Kouki Yanazawa[1], Koichi Watanabe[3], Tohru Takemasa[3], Yoshinori Takeuchi[4], Yuichi Aita[4], Naoya Yahagi[4], Yasuko Yoshida[5], Tomoaki Kuji[1], Nanami Sekine[1], Kaoru Takeuchi[6], Haruna Ueda[6], Yasushi Kawakami[2] and Kazuhiro Takekoshi[2]

[1] Graduate School of Comprehensive Human Sciences, University of Tsukuba, Tsukuba, Japan
[2] Laboratory of Laboratory/Sports medicine, Division of Clinical Medicine, Faculty of Medicine, University of Tsukuba, Tsukuba, Japan
[3] Faculty of Health and Sport Sciences, University of Tsukuba, Tsukuba, Japan
[4] Nutrigenomics Research Group, Faculty of Medicine, University of Tsukuba, Tsukuba, Japan
[5] Department of Medical Technology, Faculty of Health Sciences, Tsukuba International University, Tsuchiura, Japan
[6] Laboratory of Environmental Microbiology, Division of Basic Medicine, Faculty of Medicine, University of Tsukuba, Tsukuba, Japan
[*] These authors contributed equally to this work.

Corresponding author
Kazuhiro Takekoshi,
k-takemd@md.tsukuba.ac.jp

## ABSTRACT

**Background**. With the rapid progress of genetic engineering and gene therapy methods, the World Anti-Doping Agency has raised concerns regarding gene doping, which is prohibited in sports. However, there is no standard method available for detecting transgenes delivered by injection of naked plasmids. Here, we developed a detection method for detecting transgenes delivered by injection of naked plasmids in a mouse model that mimics gene doping.

**Methods**. Whole blood from the tail tip and one piece of stool were used as pre-samples of injection. Next, a plasmid vector containing the human erythropoietin (hEPO) gene was injected into mice through intravenous (IV), intraperitoneal (IP), or local muscular (IM) injection. At 1, 2, 3, 6, 12, 24, and 48 h after injection, approximately 50 μL whole blood was collected from the tail tip. One piece of stool was collected at 6, 12, 24, and 48 h. From each sample, total DNA was extracted and transgene fragments were analyzed by Taqman quantitative PCR (qPCR) and SYBR green qPCR.

**Results**. In whole blood DNA samples evaluated by Taqman qPCR, the transgene fragments were detected at all time points in the IP sample and at 1, 2, 3, 6, and 12 h in the IV and IM samples. In the stool-DNA samples, the transgene fragments were detected at 6, 12, 24, and 48 h in the IV and IM samples by Taqman qPCR. In the analysis by SYBR green qPCR, the transgene fragments were detected at some time point in both specimens; however, many non-specific amplicons were detected.

**Conclusions**. These results indicate that transgene fragments evaluated after each injection method of naked plasmids were detected in whole-blood and stool DNA samples. These findings may facilitate the development of methods for detecting gene doping.

## INTRODUCTION

Doping is an act of enhancing competitive abilities to achieve success by using substances or methods prohibited in sports (*JADA, 2019*). Doping in sports, particularly in festivals such as the Olympic Games and world or local championships for various competitions, is illegal and against the spirit of the game. The World Anti-Doping Agency (WADA) was established in 1999, and has been involved in scientific research on doping, anti-doping education, development of anti-doping strategies, and monitoring of the World Anti-Doping Code (hereafter, the Code) (*WADA, 2019*) to ensure the soundness and fairness of competitive sports worldwide.

With the rapid progress of genetic engineering technology and gene therapy, the WADA has been strongly alerted against gene doping. Since its early days, "gene doping" has been on the list of prohibited actions by WADA. In 2004, the WADA created a panel of experts on gene doping to investigate the latest advances in the field of gene therapy and methods for detecting doping (*WADA, 2016*). In January 2018, the WADA extended the ban on gene doping to include all forms of gene editing. Therefore, the list of prohibited substances currently includes "gene editing agents designed to alter genome sequences and/or the transcriptional or epigenetic regulation of gene expression" (*WADA, 2018*). Numerous gene editing systems have been established, including virus vectors (68.3%), CRISPR/CAS9 (0.4%), and short interfering RNA (0.2%), among others. In these systems, naked plasmids (15.4%) are used for gene therapy (*Gene therapy Clinical Trials, 2019*). They are safe compared to virus vectors and are most commonly used in gene therapy clinical trials because they do not cause infection. Naked plasmids have been used in gene therapy to treat human diseases such as obstructive arteriosclerosis and hemophilia (*Shigematsu et al., 2010*; *Roth et al., 2001*). In, generally, naked plasmids in vivo were hardly transferred to targeting cells and organs. However, recently, naked plasmids induced technique is improving (*Hu et al., 2017*; *Ito et al., 2019*; *Kanemura et al., 2008*). Additionally, gene doping and gene therapy are linked in terms of enhancing or silencing of gene expression, resulting in alterations in biological functions. Thus, they may be used for gene doping. Additionally, there are no established standard methods for detecting or preventing gene doping by naked plasmids.

In recent years, functional genes that enhance exercise performance were reported. Particularly, human erythropoietin (hEPO) was brought to the attention of the WADA. hEPO is a glycoprotein hormone mainly produced by peritubular fibroblasts in the kidney and act on bone marrow to stimulate the production of red blood cells (*Jelkmann, 2011*) Red blood cells can deliver oxygen throughout the body, and thus increase exercise performance (*Haile et al., 2019*). Endogenous hEPO production is strongly regulated by hypoxia (*Jelkmann, 2011*; *Baker & Parise, 2016*). A previous study showed that high-altitude training increases hEPO levels and improves exercise capacity by increasing red blood cell and hematocrit levels (*Park et al., 2016*). Therefore, the WADA added hEPO
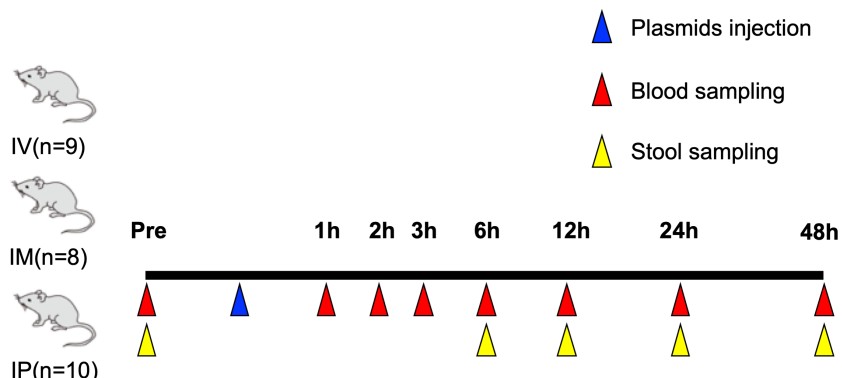

**Figure 1** Overview of animal experiments carried out in this study.

to the prohibited doping list as peptide hormones, growth factors, related substances, and mimetics. While a method is available for evaluating exogenous hEPO, there is no established method for detecting increased production of endogenous hEPO by gene editing. Therefore, we developed a detection method for detecting transgenes delivered by injection of naked plasmids in a mouse model that mimics gene doping.

## MATERIALS & METHODS

### Animal experiments

Animal experiments in this study were approved by the Animal Care Committee, University of Tsukuba (approval numbers: 19-163 and 19-425). An overview of the experiments is shown in Fig. 1. Four-week-old C57BL/6 male mice, weighing 10.3–13.9 g, were purchased from the Central Laboratories for Experimental Animals (Tokyo, Japan). The mice were bred and maintained in an air-conditioned animal house under specific pathogen-free conditions and subjected to 12/12—h light and dark cycles. The mice were fed standard mice pellet and water ad libitum. When the mice were 5 weeks of age, 50–100 µL whole blood was collected from the tail tip from an approximately one mm cut into a microtube containing EDTA 2Na while the mice were under inhalation-general anesthesia by isoflurane. One piece of stool was also collected from the mice. The mice were returned to normal breeding after blood collection. Blood and stool samples were stored at −20 °C until further analysis. After 3 weeks, when the mice were aged 8 weeks, plasmids containing hEPO gene were injected as described below.

### Naked plasmids injection and collection of samples (blood and stool)

Plasmid of pcDNA3.1(-) containing hEPO gene without introns derived from cytomegalovirus-promoter (phEPO) was purchased from GenScript Biotech Corp. (Nanjing, China). This plasmid was injected into the mice by intravenous (IV), intraperitoneal (IP), or local muscular (IM) injection. A total of 50 µg plasmid was injected into each mouse (volume; 50 µL, concentration; 1,000 ng/µL). Approximately

**Table 1** Primer sequences used in this study.

| Methods | Targets | | Sequences | Predicted size (bp) |
|---|---|---|---|---|
| qPCR Taq man probe assay | Human Erythropoietin | Forward | CGA GAA TAT CAC GAC GGG CT | 126 |
| | | Probe | 56FAM/TATCACTGT/ZEN/CCCAGACACCAAAGTT/3IABkFQ/ | |
| | | Reverse | CAG ACT TCT ACG GCC TGC TG | |
| qPCR: SYBR green assay | Human Erythropoietin | Forward | CGA GAA TAT CAC GAC GGG CT | 126 |
| | | Reverse | CAG ACT TCT ACG GCC TGC TG | |

50-µL blood samples were collected at 1, 2, 3, 6, 12, 24, and 48 h after injection and 1 piece of stool sample was collected at 6, 12, 24, and 48 h after injection.

## DNA extraction and sample preparation for qPCR assays

Total DNA was extracted from collected whole blood and the piece of stool. A NucleoSpin Blood kit (Takara Bio, Shiga, Japan) was used to isolate DNA from 50–100 µL of whole-blood. A phenol/chloroform/isoamyl alcohol solution (Nacalai Tesque, Kyoto, Japan) was used to isolate DNA from the stool sample. The concentration of total extracted DNA was measured with a NanoDrop spectrometer (Thermo Fisher Scientific, Waltham, MA, USA), and the final concentration was adjusted to 10 ng/ µL using elution buffer or distilled water (DW) for whole blood and stool-DNA. As a negative control, only elution buffer or DW was prepared.

## Primer design

To detect transgene fragment by phEPO, forward and reverse primers for the hEPO gene for both Taqman qPCR and SYBR Green qPCR assays were designed including exon-exon junctions within exon 2–4 to prevent amplification of genomic DNA from both mouse and human and ensure specificity for phEPO for in silico PCR analysis. For Taqman qPCR, primers were designed to include double quenching systems. The primers and probe were synthesized by Integrated DNA Technologies (Coralville, IA, USA) and the sequences are shown at Table 1.

### Real-time quantitative PCR (qPCR) for trans-gene fragments

PrimeTime Gene Expression Master Mix (Integrated DNA Technologies) reagent was used to perform Taqman qPCR. The template volume, primer pairs, and probe concentrations were 2 µL, 100 nM, and 50 nM, respectively, for a total reaction volume of 10 µL per sample. KAPA SYBR® FAST qPCR Master Mix (KK 4602, Kapa Biosystems, Wilmington, MA, USA) reagent was used to perform SYBR Green qPCR. The template volume and primer pair concentrations were 2 µL and 100 nM, respectively, for a total reaction volume of 10 µL per sample. The phEPO concentration was 10 pg/µL ($1.54 \times 10^9$ copy/µL) for use as a standard material to perform absolute quantification of the copy number of trans-gene fragments. As negative controls, 2 µL of DW was used. The conditions maintained in the thermal cycler were 95 °C for 20 s; and 95 °C for 1 s and 60 °C for 20 s, for 35 cycles in the Taqman probe assay and 95 °C for 20 s; 95 °C for 30 s, and 60 °C for 30 s for 35 cycles

followed by melting curve analysis in the SYBR Green assay in a QuantStudio 5 Real-Time PCR System (Thermo Fisher Scientific). All samples were measured in duplicate, and the coefficient of determination (R2) of the standard curve was equal to 0.98.

### Specificity and accuracy of Taqman qPCR assay

To confirm the accuracy and specificity of the Taqman qPCR assay for trans-genes introduced by hEPO, blood-DNA and stool-DNA samples at each time point were pooled; negative control DNA samples were prepared using genomic DNA isolated by phenol chloroform extraction from 5 human cell lines: human renal tubular cells (HK-2 cells: CRL-219, ATCC, Manassas, VA, USA), human embryonic fibroblasts (HEF cells: JCRB 1006.7, JCRB Cell Bank; original developers: Kouchi and Namba), human embryonic kidney cells (HEK293 cells: Thermo Fisher Scientific), Human hepato cellular carcinoma cell (HuH-7 cells: JCRB0403, JCRB Cell Bank; original developers: Nakabayshi and Sato), and human mesenchymal stem cells (HMS cells: UE6E7T-2; JCRB 1133, original developer: Umezawa, A.). The concentration of the negative control DNA samples was adjusted to 10 ng/μL. These DNA samples were subjected to Taqman qPCR assay again as described above. After Taqman qPCR, the amplicons were subjected to electrophoresis in a 2% agarose gel and visualized by staining with ethidium bromide to confirm the amplicon size. The remaining amplicons of blood-DNA were cleaned up using NucleoSpin Gel and a PCR Clean-up kit (Takara Bio). Because amplicons of non-specific sizes were detected in the stool-DNA after Taqman qPCR, the remaining amplicons were subjected to electrophoresis in a 2% agarose gel and visualized; amplicons of the target size were cut from the gel and their DNA was extracted using NucleoSpin Gel and PCR Clean-up kit. The cleaned amplicon-DNA were subjected to Sanger sequencing performed by FASMAC, Inc. (Kanagawa, Japan). The sequencing data were analyzed using CLC Sequence Viewer ver. 8.0 and BioEdit Ver. 7.0.5.3 to determine whether the amplicon sequences matched the transgene of phEPO.

### Statistics

Data are shown as the mean ± SEM. All data were subjected to Kruskal–Wallis H test (one-way analysis of variance of ranks), followed by a two-stage Benjamini, Krieger, and Yekutieli false discovery rate procedure as a post-hoc test using GraphPad Prism version 7.04 (GraphPad, Inc., La Jolla, CA, USA). A $p$ value less than 0.05 was considered as statistically significant.

## RESULTS

### hEPO transgene was detected in two different samples by qPCR

In the blood specimens, using the TaqMan probe, transgene fragments were detected at 1, 2, 3, 6, and 12 h in the IV and IM groups and at 1, 2, 3, 6, 12, and 48 h in the IP group. In contrast, in the SYBR green assay, they were detected at 1, 2, 3, and 6 h in the IV and IM groups, but were not detected in the IP group. In the stool specimens, using the TaqMan probe, they were detected at all time points in the IV and IM groups and at 6 and 12 h in the IP group. In contrast, using SYBR green, the trans-gene fragments were not detected (Tables 2 and 3).
**Table 2** Detection of transgene fragments in blood samples by TaqMan qPCR and SYBR Green qPCR.

| Method | Group | Copy/ µL of Transgene | | | | | | | |
|---|---|---|---|---|---|---|---|---|---|
| | | **Pre** | **1 h** | **2 h** | **3 h** | **6 h** | **12 h** | **24 h** | **48 h** |
| Taqman qPCR | IV | 0 | 739,964.7[**] ±423,342.3 | 27,929.9[**] ±6,674.9 | 53,345[**] ±23,608.6 | 9,274.3[**] ±2,449.9 | 7,948[**] ±7,408.7 | 26.5 ± 8.4 | 41.4 ± 12.8 |
| | IM | 0 | 84,334[**] ±12,302.6 | 21,906.5[**] ±7,682.8 | 58,617.3[**] ±15,427.5 | 14,651.9[**] ±4,555.5 | 209.8[**] ±48.1 | 5.5 ± 1.5 | 48.7 ± 18.4 |
| | IP | 0 | 135,428.1[**] ±71,292.2 | 2,655.3[**] ±1,697.2 | 2,553.3[**] ±1,266.9 | 532.2[**] ±318.7 | 611.6[**] ±542.5 | 5.3 ± 0.9 | 337.7[**] ±247.4 |
| SYBR qPCR | IV | 34.9 ± 4.6 | 42,3512.2[**] ±24,9442.4 | 3,424.8[**] ±963.9 | 15,471.8[**] ±8,637.2 | 2,249.8[**] ±634.1 | 1,455.3 ±1,370.1 | 45.4 ± 6.9 | 23.8 ± 2.2 |
| | IM | 35.8 ± 2.4 | 31,633.2[**] ±5,109.2 | 3,114[*] ±1,340.4 | 16,818 ±4,902.1 | 4,397.9[*] ±1,546.8 | 44.9 ± 8.4 | 25.7 ± 1.2 | 28.7 ± 3.1 |
| | IP | 53.4 ± 3.9 | 71,880.8 ±44,500.8 | 235.3 ±152.1 | 441.6 ±217.9 | 107.9 ±57.9 | 74.6 ± 53.3 | 27.6 ± 1.7 | 54.1 ± 25.8 |

**Notes.**

Transgene fragments were detected in all samples in TaqMan qPCR.

IV, intravenous injection; IM, intramuscular injection; IP, intraperitoneal injection of the hEPO coding naked plasmid.

[**]$p < 0.01$ vs Pre in each group.

[*]$p < 0.05$ vs Pre in each group.
**Table 3 Detection of transgene fragments in stool samples by TaqMan qPCR and SYBR Green qPCR.**

| Method | Group | Copy/μL of Transgene | | | | |
|--------|-------|------|------|------|------|------|
| | | pre | 6 h | 12 h | 24 h | 48 h |
| Taqman qPCR | IV | 0 | $30.0 \pm 11.4$[**] | $85.4 \pm 39$[**] | $967.5 \pm 388.3$[**] | $10.0 \pm 1.4$[**] |
| | IM | 0 | $15.1 \pm 4.9$[**] | $85.8 \pm 30.8$[**] | $913.9 \pm 263$[**] | $43.2 \pm 12.2$[**] |
| | IP | 0 | $1,809.6$[**] $\pm 1,587.9$ | $180.8 \pm 148.1$[**] | 0 | 0 |
| SYBR qPCR | IV | $426.9 \pm 159.8$ | $65.4 \pm 16.6$ | $111.2 \pm 18.3$ | $239.9 \pm 62.5$ | $34 \pm 5.4$ |
| | IM | $762.4 \pm 239.3$ | $44 \pm 5.7$ | $117.7 \pm 23.6$ | $263.8 \pm 56.1$ | $46.1 \pm 7.1$ |
| | IP | $885.1 \pm 403.6$ | $1,111.6 \pm 965.2$ | $103.8 \pm 35$ | $110.1 \pm 10.2$ | $51.6 \pm 8.4$ |

**Notes.**

Transgene fragments were detected in all samples in TaqMan qPCR.

IV, intravenous injection; IM, intramuscular injection; IP, intraperitoneal injection of hEPO coding naked plasmid.

[**] $p < 0.01$ vs Pre in each group.

[*] $p < 0.05$ vs Pre in each group.

### TaqMan qPCR specifically and accurately detected the transgene

After confirming the specificity and accuracy of the Taqman qPCR assay for detecting trans-gene fragments, a specific amplification curve was detected in pooled blood-DNA and stool-DNA samples, while no amplification was observed in any human cell lines or DW as the negative controls (Fig. 2). In agarose gel electrophoresis, a single band for the amplicon was detected in all whole blood-DNA samples by Taqman qPCR (Fig. 3). In contrast, stool-DNA samples showed two bands for the amplicon, indicating that one was the hEPO gene and the other was a non-specific amplicon. No amplicon band was detected in any of the human cell lines or DW as negative controls (Fig. 4). Further analysis to confirm the specificity and accuracy of Taqman qPCR was conducted by Sanger sequencing. In the whole blood samples, a separated single peak was detected in the IV, IM, and IP specimens (Fig. 5). In the stool samples, no single peak was detected in any groups (Fig. 4). To confirm the specificity and accuracy of the amplicon, Sanger sequencing was conducted. In whole blood samples, separated and single peak sequences were detected in the IV, IM, and IP specimens. In the stool samples, no single sequence was detected in any group. Concordances between hEPO-reference and amplicon sequences in all samples were high (Figs. 4 and 5).

### DISCUSSION

In recent years, gene editing systems have been improved and developed for gene therapy and gene therapy for various diseases is being evaluated in clinical trials (ex. critical limb ischemia, soft tissue sarcomas, retinal dystrophy, muscular dystrophy) (*Shigematsu et al., 2010*; *Xiao et al., 2017*; *Russell et al., 2017*; *Mendell et al., 2015*). However, with the development of gene therapy technology, the risk of gene doping is increasing. Particularly, hEPO is a target gene for gene doping because it may improve endurance capacity and exercise performance (*Haile et al., 2019*; *Durussel et al., 2013*). An hEPO formulation has been developed for doping which shows a different molecular weight from endogenous hEPO, enabling its detection (*Salamin et al., 2018*). However, detection remains difficult

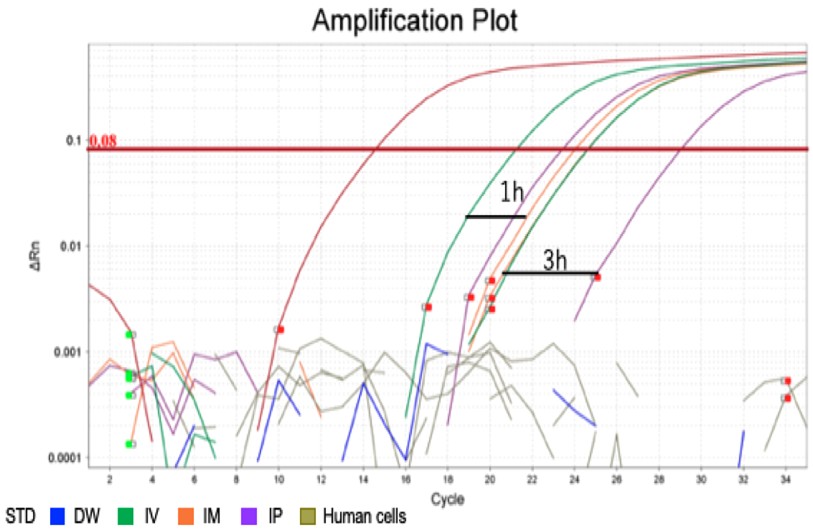

**Figure 2 Amplification plot of blood sample and human cells.** To confirm the accuracy and specificity of the Taqman qPCR assay for trans-genes introduced by hEPO, blood-DNA and stool-DNA samples at each time point were pooled and Taqman qPCR was conducted. Specific amplification was detected in blood samples. There were no amplification in human cells and distilled water (DW).

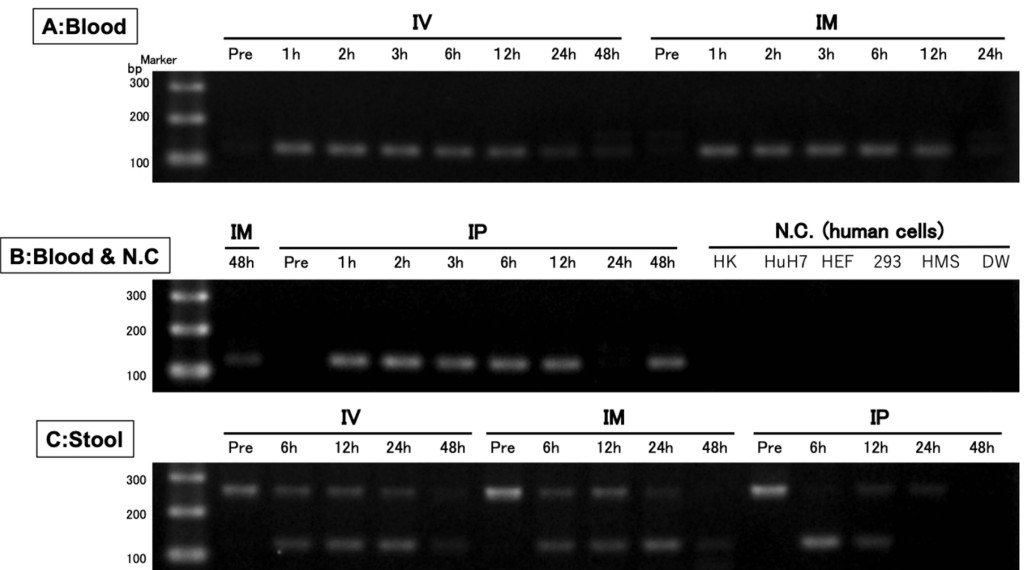

**Figure 3 Confirmation of the accuracy and specificity in TaqMan qPCR using agarose gel electrophoresis.** A single band was detected from blood samples, whereas two bands were detected from stool samples. There was no band in the negative control (human cells). IV is intravenous injection, IM is intramuscular injection, and IP is intraperitoneal injection of hEPO coding naked plasmid. HK: HK-2 cells (10 ng/μL), HuH7: HuH-7 cells (10 ng/μL), HEF: Human embryonic fibroblasts cells (10 ng/μL), 293: Human embryonic kidney 293 cells (10 ng/μL), HMS: Human mesenchymal stem cells (10 ng/μL), DW: Distilled water.

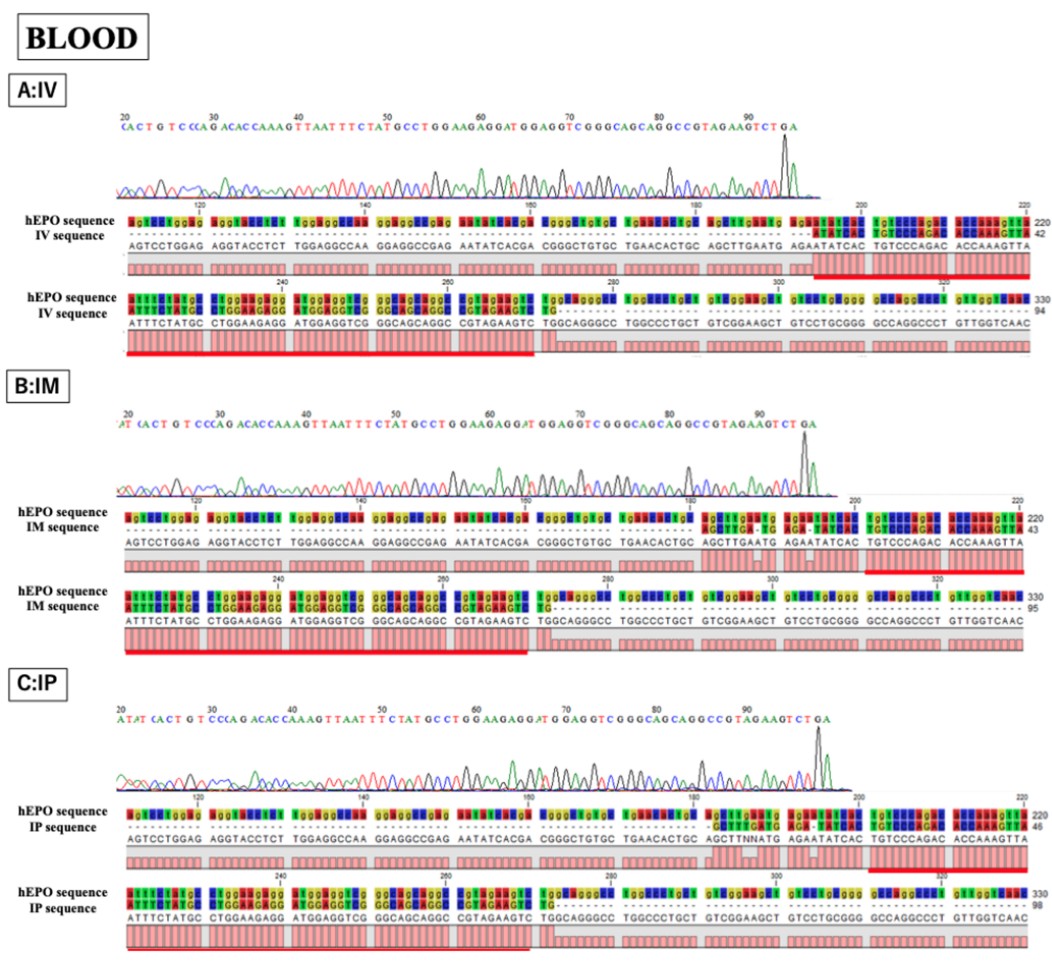

**Figure 4 Confirmation of the accuracy and specificity of TaqMan qPCR from blood samples using Sanger sequence methods.** Single peak was detected from all samples and concordances between hEPO-reference and amplicon sequences in all samples was high. IV is intravenous injection, IM is intramuscular injection, and IP is intraperitoneal injection of hEPO coding naked plasmid. Red lines indicatethe matching sequences between hEPO-reference and amplicon sequences.

because there is no difference between the genome product and transgene product. Thus, we aimed to detect the transgene in a gene doping mouse model using naked plasmids.

We developed a method for detecting the trans-gene delivered by a naked plasmid coding for hEPO and examined the differences in detection sensitivity depending on the route of administration, as various organs may be targeted by the gene. We detected the transgene fragment following IV, IP, or IM injection by two qPCR methods and TaqMan assay. Tozaki et al. (2018) reported that plasmid injection into the muscle could be detected by droplet digital PCR (ddPCR) until 2–3 weeks after injection in micro mini pigs. In contrast, hEPO transgene fragments were detected in IP blood samples until 48 h after injection in this study. There are three reasons for this difference: (i) the injection amount was lower than that used in the previous study (previous: 250 μg, this study: 50 $\mu$g); (ii) species differences (metabolic speed can differ because of enzyme activity differences); and (iii)

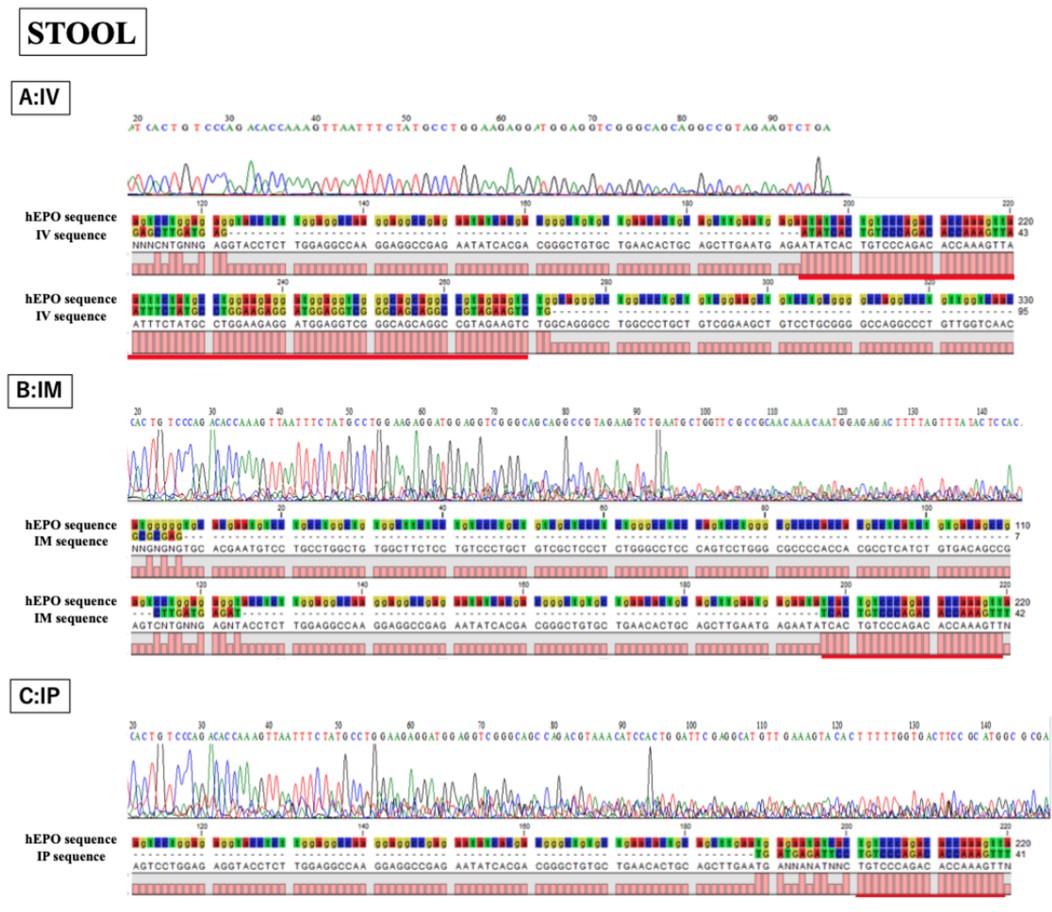

**Figure 5 Confirmation of accuracy and specificity in TaqMan qPCR from stool samples using Sanger sequence methods.** Single peak was detected only in IV sample, but concordance between hEPO-reference and amplicon sequences in all samples was confirmed. IV is intravenous injection, IM is intramuscular injection, and IP is intraperitoneal injection of hEPO coding naked plasmid. Red lines indicatethe matching sequences between hEPO-reference and amplicon sequences.

the PCR method and sensitivity show some differences (previous: ddPCR/high sensitivity, this study: Taqman qPCR/normal). However, ddPCR is more costly than Taqman qPCR, and thus ddPCR cannot be used to analyze large numbers of samples. In this study, we compared the Taqman qPCR and SYBR Green PCR assay to detect gene doping. As a result, Taqman qPCR detected higher levels of the gene than the SYBR Green assay and for a longer period of time. Therefore, this model can be used for gene doping detection using naked plasmids.

Based on the results of qPCR, stool samples are useful for gene doping inspection because they can be collected non-invasively. The collection of blood samples is invasive; however, transgene fragments were detected from a smaller sample volume. Hence, blood and stool samples are useful for this detection.

We evaluated the accuracy of the TaqMan assay by three methods using blood and stool samples, which confirmed the presence of (i) specific amplifications in the TaqMan assay,

(ii) transgene molecular size by electrophoresis, and (iii) transgene DNA sequence by Sanger sequencing. The hEPO amplification products and DNA sequence agreed with the reference in all blood samples. In contrast, in the stool sample, several non-specific bands were observed in agarose gel electrophoresis and waveform disturbances were detected in the Sanger method. The stool sample may have contained a high level of enterobacteria DNA. Although Sanger sequencing did not provide accurate results, TaqMan qPCR was specific and thus may be adapted for transgene detection.

Primer dimers formed in SYBR Green reactions cause decrease the sensitivity of real-time PCR. This problem can be avoided by using a specific detection system with TaqMan MGB probes, where non-specific amplification is not detected because of the binding of the specific MGB probe only to a 100% complementary nucleotide sequence (*Orlando, Pinzani & Pazzagli, 1998*).

There was a limitation to this study. We did not examine protein expression levels, the number of red blood cells, or hemoglobin levels. Thus, whether the plasmid functioned after injection remains unclear and its gene-doping require further analysis. However, we detected the transgene in blood and stool samples in a specific and accurate manner. These results will contribute to the development of gene doping detection methods.

## CONCLUSIONS

In the present study, a transgene fragment was detected in whole blood DNA and stool DNA samples by qPCR. Particularly, TaqMan qPCR is specific and accurate compared to the SYBR Green assay. Additionally, stool and whole blood sample collection is relatively non-invasive and can be used for gene doping inspection. These results may lead to the development of standard detection methods for gene doping and deter the use of gene doping by athletes.

## ACKNOWLEDGEMENTS

We thank Katsuyuki Tokinoya for commenting on this study design and data analysis.

### Funding
This work was supported by a grant from the promotional business of doping prevention activities, Japan Sports Agency (JSA). The funders had no role in study design, data collection and analysis, decision to publish, or preparation of the manuscript.

### Grant Disclosures
The following grant information was disclosed by the authors:
Japan Sports Agency (JSA).

### Competing Interests
The authors declare there are no competing interests.

## Author Contributions

- Kai Aoki, Takehito Sugasawa and Kouki Yanazawa conceived and designed the experiments, performed the experiments, analyzed the data, prepared figures and/or tables, authored or reviewed drafts of the paper, and approved the final draft.
- Koichi Watanabe and Kazuhiro Takekoshi conceived and designed the experiments, authored or reviewed drafts of the paper, and approved the final draft.
- Tohru Takemasa, Yoshinori Takeuchi, Yuichi Aita, Yasuko Yoshida, Tomoaki Kuji, Nanami Sekine, Haruna Ueda and Yasushi Kawakami analyzed the data, authored or reviewed drafts of the paper, and approved the final draft.
- Naoya Yahagi and Kaoru Takeuchi performed the experiments, authored or reviewed drafts of the paper, and approved the final draft.

## Animal Ethics

The following information was supplied relating to ethical approvals (i.e., approving body and any reference numbers):

Animal experiments in this study were approved by the Animal Care Committee, University of Tsukuba (approval numbers: 19-163 and 19-425).

## Data Availability

The raw measurements are available in the Supplementary Files.

## Supplemental Information

Supplemental information for this article can be found online at http://dx.doi.org/10.7717/peerj.8595#supplemental-information.

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
