# Peer review of "The detection of trans gene fragments of hEPO in gene doping model mice by Taqman qPCR assay"

_PeerJ, doi:10.7717/peerj.8595_

## Round 0.1 · original submission · Minor Revisions

Please read and address the comments made by the reviewers, trying to incorporate their suggestions as much as possible. Please specially, make sure to address reviewer 2 comments (their comments are in the attached PDF). Make sure to correct both grammatical and spelling errors, especially in the names of the supplemental files. Once I receive your corrected manuscript, I will re-review it myself before acceptance

·

Basic reporting

The manuscript is well written. Introduction covered all the content needed to understand the results with literature well referenced and current. All figures are in good quality, however, it is necessary to improve the subtitles according to the journal standard and add relevant information for the understanding of the images and graphics presented (see General comments).

Experimental design

This work is within Scope of the journal and shows a well-designed and rigorous research with coherent methods to achieve the proposed objectives. Methods were described with sufficient detail and information to replicate.

Validity of the findings

This work contributes with many information to understanding and development methods for gene doping and it has scientific relevance in the research field. My opinion is that the score assigned to the manuscript is good for publication in this jornal. My concerns and suggestions are described in General comments.

Additional comments

Line 188: The authors described "Transgene fragments were detected in all samples in TaqMan qPCR", but in results section (lines 179-180) was described "In the blood specimens, using the TaqMan probe, transgene fragments were detected at 1, 2, 3, 6, and 12 h in the IV and IM groups and at 1, 2, 3, 6, 12, and 48 h in the IP group". Why were samples at times 24 and 48 h not considered?

Line 190: Statistical analysis were presented in the tables legend 2 and 3 "**: p < 0.01 vs Pre in each group. *: p < 0.05 vs Pre in each group", but this information is nowhere in the tables.

Table 2: Why was the detection of transgene fragments in blood samples by TaqMan qPCR and SYBR Green qPCR higher in 3h than 2h? Because according to the results, the tendency of values is to decrease in blood samples. These results were obtained from how many independent experiments? This information is missing from material and methods.

In line 195 is written " Transgene fragments were detected in all samples in TaqMan qPCR" but was not in accordance with the results given in table 3 for IP samples (24 and 48 h).

Lines 208 and 210: Check and correct the figures citations.

Lines 219-220, 225-226, 236-240, and 251-255: In the figures legend (2, 3, 4, and 5) were described results, which already were presented in the images. It would be more useful to add important information to understand how that result presented in the image was obtained. This information makes reading more enjoyable and practical, as the reader does not have to go back to the material and methods to check.

Figure 2: The results presented for negative controls and specific amplification are clear, but the times of 1 and 3h inserted in the image are confusing. Was the 3h time only used for the IP sample?

Figure 4 and Figure 5: The hEPO-reference and amplicon sequences identifications are exchanged in the images.

Reviewer 2 ·

Basic reporting

Cited references were insufficient to provide the scientific background.
In raw date, several cells were written in Chinese and Japanese.

Experimental design

It is not suitable for animal model in this study.

Validity of the findings

no comment

Annotated reviews are not available for download in order to protect the identity of reviewers who chose to remain anonymous.

---

## Round 0.2 · accepted · Accept

Thanks for submitting your work to PeerJ. Your manuscript is obviously controversial and not all reviewers were convinced of the importance of your contribution. Given the lack of strong experimental arguments by reviewers against the work you have presented, and given PeerJ's criteria, I have decided to accept your work. I thank you again for submitting your work to PeerJ.

Reviewer 2 ·

Basic reporting

It is not problems in English.

Several references is not suitable to introduce the research background of simply injection methods of naked plasmid DNAs. For example, in "Ito T et al., Mol Pharm. 2019", they used several excipients as non-viral gene carriers, and ,in Kanemura H et al., Hepatol Res. 2008, they performed gene transfer under hypertonic stress similar to the hydrodynamic injection.

Figures, tables and raw data is enough to publish.

Experimental design

Not enough. The authors should establish gene doping mice model and investigate the effect of gene doping using that model.

Validity of the findings

The novelty of this research is poor. Because the clearance of naked plasmid DNA from the body is well-investigated since 1990s.

Additional comments

no comment